

# VOE: automated analysis of variant epitopes of SARS-CoV-2 for the development of diagnostic tests or vaccines for COVID-19

Danusorn Lee[1] and Unitsa Sangket[1,2]

[1] Division of Biological Science, Faculty of Science, Prince of Songkla University, Hat Yai, Songkhla, Thailand
[2] Center for Genomics and Bioinformatics Research, Faculty of Science, Prince of Songkla University, Hat Yai, Songkhla, Thailand

## ABSTRACT

**Background**. The development of serodiagnostic tests and vaccines for COVID-19 depends on the identification of epitopes from the SARS-CoV-2 genome. An epitope is the specific part of an antigen that is recognized by the immune system and can elicit an immune response. However, when the genetic variants contained in epitopes are used to develop rapid antigen tests (Ag-RDTs) and DNA or RNA vaccines, test sensitivity and vaccine efficacy can be low.

**Methods**. Here, we developed a "variant on epitope (VOE)" software, a new Python script for identifying variants located on an epitope. Variant analysis and sensitivity calculation for seven recommended epitopes were processed by VOE. Variants in 1,011 Omicron SRA reads from two variant databases (BCFtools and SARS-CoV-2-Freebayes) were processed by VOE.

**Results**. A variant with HIGH or MODERATE impact was found on all epitopes from both variant databases except the epitopes KLNDLCFTNV, RVQPTES, LKPFERD, and ITLCFTLKRK on the S gene and ORF7a gene. All epitope variants from the BCFtools and SARS-CoV-2 Freebayes variant databases showed about 100% sensitivity except epitopes APGQTGK and DSKVGGNYN on the S gene, which showed respective sensitivities of 28.4866% and 6.8249%, and 87.7349% and 71.1177%.

**Conclusions**. Therefore, the epitopes KLNDLCFTNV, RVQPTES, LKPFERD, and ITLCFTLKRK may be useful for the development of an epitope-based peptide vaccine and GGDGKMKD on the N gene may be useful for the development of serodiagnostic tests. Moreover, VOE can also be used to analyze other epitopes, and a new variant database for VOE may be further established when a new variant of SARS-CoV-2 emerges.

# INTRODUCTION

On December 31, 2019, the World Health Organization (WHO) was notified of cases of pneumonia of unknown cause in Wuhan City, China. On January 7, 2020, a novel coronavirus was identified as the cause by Chinese authorities and tentatively designated

Corresponding author
Unitsa Sangket, unitsa.s@psu.ac.th

2019-nCoV (*World Health Organization, 2022b*). Mutations in the severe acute respiratory syndrome coronavirus 2 (SARS-CoV-2) genome resulted in a novel variant coronavirus (*World Health Organization, 2022a*). Coronavirus variants of concern (VOCs) are variants of global public health significance that can be associated with one or more of the following changes: an increase in transmissibility or adverse change in COVID-19 epidemiology; an increase in virulence or change in the clinical disease pattern; or a decrease in the effectiveness of public health and social interventions, diagnostics, vaccines, or therapeutics. The WHO has reported five VOCs of SARS-CoV-2 that increase transmissibility or virulence symptoms, or decrease diagnostic efficacy. The VOCs are known as Alpha, Beta, Gamma, Delta and Omicron (*World Health Organization, 2022a*).

The Alpha VOC, variant B.1.1.7 detected by the plaque-reduction neutralization test (PRNT50), reduced the neutralizing activity of the BNT162b2-Elicted serum (*Liu et al., 2021*). The Delta VOC, B.1.617.2, was reported after the efficacy of the COVID-19 vaccine BNT162b2 ChAdOx1 was seen to diminish after the receipt of one dose (*Lopez Bernal et al., 2021*). The Omicron variant, B.1.1.529, was first identified on November 25, 2021, in Gauteng Province, South Africa (*Callaway, 2021*). The Omicron variant contains mutations that suggest it may be more infectious and transmissible and better able to evade innate immunity and neutralize antibody activity than the wild-type virus (*Centre of Disease Prevention E, 2021*; *Ukhsa, 2022*; *Chen et al., 2022*). Following the emergence of the original B.1.1.529 Omicron variant, several subvariants of Omicron have emerged, including BA.1, BA.2, BA.3, BA.4, and BA.5 (*Yao et al., 2022*). New subvariants of Omicron continue to emerge (*Vitiello et al., 2022*). In the United Kingdom in January 2022, XE, a new BA.1–BA.2 recombinant was isolated (*Mohapatra et al., 2022*). In August 2022, an Omicron subvariant XBB, a recombinant of the BA.2.10.1 and BA.2.75 sublineages (*World Health Organization, 2022c*) was detected. Currently circulating variants of interest (VOIs) of Omicron reported by the WHO are XBB.1.5 and XBB.1.16 (both recombinants of the BA.2.10.1 and BA.2.75 sublineages, the former with a breakpoint in S1 protein) and EG.5 (a descendant of XBB.1.9.2) (*World Health Organization, 2023*).

Rapid diagnostic tests (RDTs) are used to diagnose SARS-CoV-2 in external laboratories. RDTs are easy to use, provide rapid results and are less expensive than nucleic acid amplification test (NAATs) such as rRT-PCR, but they are also generally less sensitive and specific and must be confirmed with an NAAT. There are two types of RDT—antigen (Ag) RDTs and antibody (Ab) RDTs (*World Health Organization, 2020a*). However, the WHO recommends that Ab-RDTs should not be used to identify active infections in clinical care or for contact tracing (*World Health Organization, 2020b*). Ab-RDTs may be useful for sousveillance studies to aid in the investigation of an ongoing outbreak or the retrospective assessment of the rate or extent of an outbreak. Ag-RDTs primarily detect the nucleocapsid of the virus in respiratory secretions. Ag-RDTs detect antigens from clinical specimens using an immunochromatographic assay format (*World Health Organization, 2020a*). The test kit typically consists of a nitrocellulose strip enclosed in a plastic cassette with a sample well. The sample is mixed with test buffer and placed in the sample well. Target antigens in the sample bind to a labeled antibody and they migrate along the test strip together. They are then captured by a second antibody attached to the test strip, causing a detectable

 

color change (*World Health Organization, 2020c*; *Kupferschmidt, 2021*; *Peto et al., 2021*). Ag-RDTs may have product design or quality issues such as insufficient antibody quantity or insufficient affinity for the target antigen, and possible cross-reactivity with other microorganisms (*World Health Organization, 2021*). The emergence of future VOCs of this β-coronavirus with an altered mutation pattern in the nucleocapsid protein will require the re-evaluatation of the performance of Ag-RDTs (*Osterman et al., 2022*). With regard to vaccines, results suggest that vaccine efficacy is much lower against the symptomatic disease caused by the Omicron variant than the Delta variant. Two doses of the ChAdOx1 nCoV-19 vaccine showed no protective effect against infection by the Omicron variant 20 to 24 weeks after the second dose (*Andrews et al., 2022*). Vaccines against SARS-CoV-2 are typically DNA or RNA vaccines. Alternative epitope-based vaccines can be developed by using immunoinformatics to identify epitopes or protein regions that are physiologically important to the virus and can elicit an immune response (*Ayra & Arora, 2020*; *Fahmi et al., 2021*). Immunoinformatics has been used predict peptide-MHC complexes, and comparative molecular docking analyses have led to the identification of potential peptides for peptide vaccine development (*Waqas et al., 2020*). The predicted epitopes RVQPTES, APGQTGK, DSKVGGNYN, and LKPFERD (*Ferreira et al., 2021*) were highly antigenic for the development of an epitope-based peptide. The predicted epitopes KLNDLCFTNV and ITLCFTLKRK (*Can et al., 2020*) were ideal antigens for the development of an epitope-based peptide, and GGDGKMKD (*Can et al., 2020*) was ideal for a serodiagnostic assay. However, the significant predicted epitopes should be further examined in wet lab studies (*Can et al., 2020*; *Ansori et al., 2021*).

Epitope APGQTGK is part of epitope S406-420 located on the SARS-CoV-2 spike protein. Epitope DSKVGGNYN is part of epitope S439–454 and epitope LKPFERD is part of epitope S455–469. S406-420, S439–454 and S455–469 were shown to induce production of the neutralization antibody in SARS-CoV-2 pseudovirus neutralization. The study validated the immunogenicity of the epitopes by immunizing mice and suggests that the epitopes could be used to design a broad-spectrum betacoronavirus vaccine (*Lu et al., 2021b*). Furthermore, epitope KLNDLCFTNV induced robust *in vivo* T cell responses and an IFN-γ ELSPOT assay demonstrated that the epitope may help design a vaccine against multiple virus variants (*Shen et al., 2022*). Nonspecific antibodies with an epitope of a SARS-CoV−2 VOC could affect the sensitivity of Ag-RDTs and efficacy of vaccines. The problem was illustrated by the proposal of an epitope of SARS-CoV-2 that is likely to be more specific in the antibody of Ab-RDTs and in the neutralizing antibody of a vaccine (*Can et al., 2020*). Since the variant of this epitope has not been determined, if the epitope is used to develop Ag-RDTs and DNA or RNA vaccines, the tests could be of low sensitivity and the vaccines of poor efficacy. However, bioinformatics tools can be used to transfer mutations from sequence read archive (SRA) data into a variant calling format (VCF) (*Sangket et al., 2022*; *Chanasongkhram, Damkliang & Sangket, 2023*). Currently, a SARS-CoV-2 genome of about four million samples is held in the NCBI database and approximately three million SRA runs and reference genomes and all other SARS-CoV-2 data are entered daily in the NCBI database (*National Center for Biotechnology Information, 2022*). Tools such as Bowtie2 (*Langmead & Salzberg, 2012*) and SAMtool (*Li et al., 2009*)

are based on sequence alignment and variant calling of SRA reads from SARS-CoV-2 whole genome sequencing in VCF. In addition, the SARS-CoV-2 Freebayes pipeline (*Farkas et al., 2021*) has been developed to explore SARS-CoV-2 mutations from SRA reads in VCF format and output a format suitable for variant filtering and annotation.

The previous studies (*Can et al., 2020*; *Lu et al., 2021b*; *Ansori et al., 2021*) used predicted epitopes to design epitope-based vaccines and a serodiagnosis assay. Essential useful bioinformatics entities were also reported to solve the SARS-CoV-2 conundrum (*Fahmi et al., 2021*). However, bioinformatics tools for rapid screening of variants located on an epitope do not exist. In this study, we aim to develop a "variant on epitope (VOE)" software, a new Python script for identifying variants located on an epitope for the development of Ag-RDTs and epitope-based vaccines. The requirement for this tool is based on the hypothesis that if a missense variant is found on an epitope, the sensitivity may be decreased because of the less compact binding of an epitope from the missense variant. However, if no variant or a synonymous variant is found on the epitope, epitope binding will be the same, resulting in identical sensitivity.

## MATERIALS & METHODS

### Experimental setup

All experiments in this study were performed in a 64-bit architecture server with a 1 TB SSD, 64 GB of RAM, two Intel(R) Xeon(R) Silver 4110 CPUs @ 2.10 GHz running Ubuntu 22.04.2 LTS. All analysis software was installed through the Bioconda channel, which can be downloaded and installed using the instructions in the Conda documentation.

### VOE development

The overview of the VOE development workflow in Fig. 1 summarizes the following process. To begin the process, genomic SARS-CoV-2 SRA accession numbers were selected from the NCBI SRA database filtered with paired library layout, illumina platform and 31/5/2022–31/5/2023, creating input files. Using the NCBI SRA toolkit, 1,961 SRA files were split into paired-end FASTQ files. To create intermediate files, the paired-end FASTQ files were referenced and assembled using the HaVoc pipeline, which includes a read trimming pre-processing tool (*Truong Nguyen et al., 2021*). Subsequently, 1,200 assembled SRA files were identified as belonging to the SARS-CoV-2 Omicron lineage, using Pangolin version 4.3 (*Rambaut et al., 2020*). The Omicron lineage accession number was selected. The Omicron SRA files in csv format were split into paired-end FASTQ files using the NCBI SRA toolkit (https://trace.ncbi.nlm.nih.gov/Traces/sra/sra.cgi?view=software). All sequence reads in the SRA files were trimmed to remove low-quality reads and bases, and adapter sequences removed using Fastp (*Chen et al., 2018*) with a cut-off from the SARS-CoV-2 Freebayes pipeline.

Before the next step, the trimmed sequence reads were validated. The quality of the trimmed reads of Omicron SRA files were assessed using the following four statuses of FastQC version 0.12.1 (*Simon, 2010*): (1) Per base sequence quality (median value of each base greater than 25), (2) per sequence quality (median quality greater than 27), (3) per base N content (N base less than 5% at each read position) and (4) adapter content (adapter

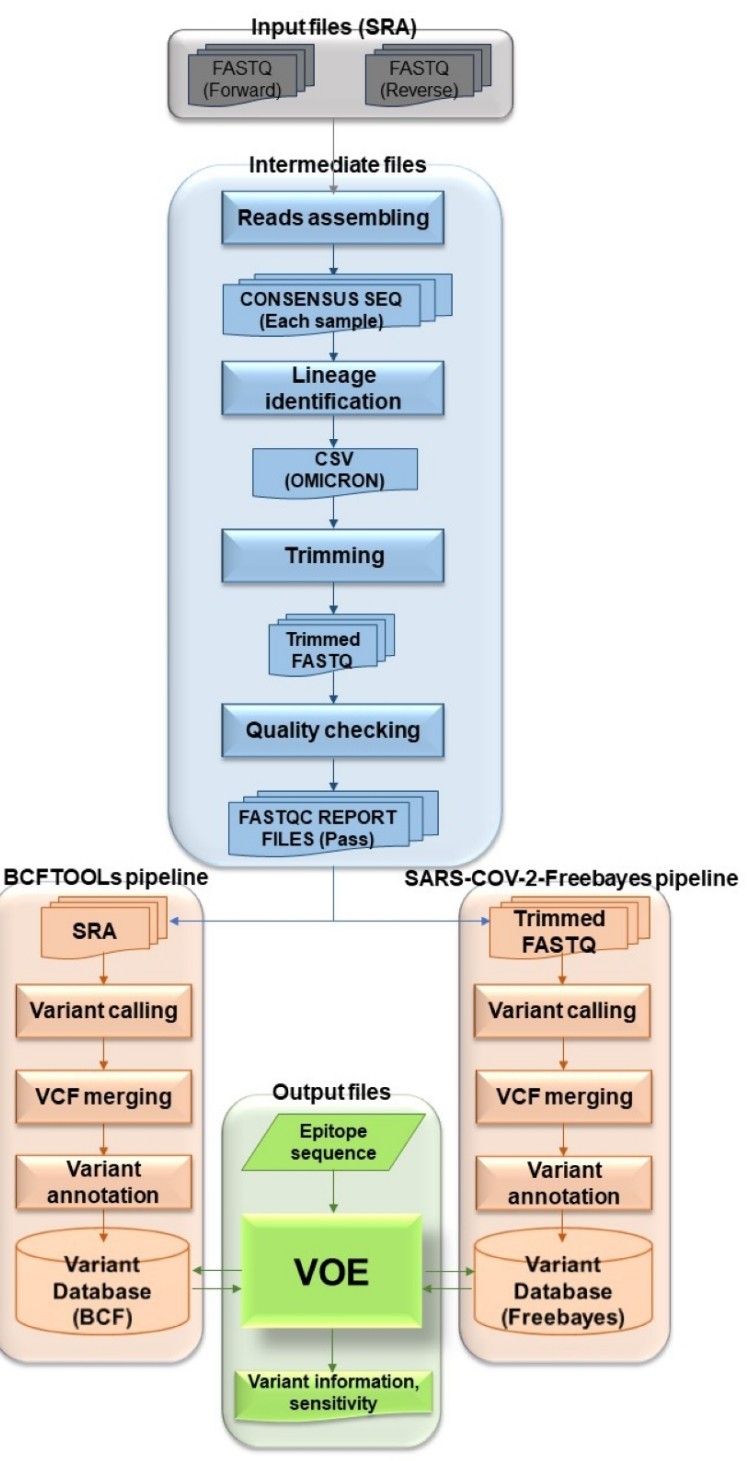

**Figure 1 Workflow describing all processes and steps in this study.**

sequences at each position less than 5% of all reads) (*Simon, 2010*). The output FastQC files were combined with MultiQC v1.4 (*Ewels et al., 2016*). Files that failed the quality assessment were removed for the variant calling step.

To create the BCFtools variant database, selected fastp (trimmed fastq) files were processed using Bowtie2 Aligner (*Langmead & Salzberg, 2012*) and BCFtools variant caller (*Li, 2011*). All reads in fastp files were aligned against Wuhan-Hu-1 (NCBI accession: NC_045512.2) using Bowtie2 version 2.3.4.1 (*Langmead & Salzberg, 2012*). The SAM files were converted to BAM files and the BAM files were sorted and indexed using SAMtools (*Li et al., 2009*). The BAM files were accessed by BCFtools version 1.17 (*Li, 2011*) as variants (read depth > 100) (*Lythgoe et al., 2021*). All VCF files from the variant calling step were merged with Jacquard version 1.1.5 (*Farkas et al., 2021*) and annotated with SnpEff version 5.0e (*Cingolani et al., 2012*). The merged, annotated file was called BCFtools variant database. For the SARS-CoV-2 Freebayes variant database, all selected SRA files were variants called one at a time from the SARS-CoV-2 Freebayes pipeline.

The raw variant databases were then ready to be modified by VOE. The modified BCFtools and SARS-CoV-2 Freebayes variant databases were presented in the format shown in Table 1. Headers and uninteresting information were deleted. POS and # CHROM were swapped. In the INFO column, the variant was annotated using the format of SnpEff version 5.0e. Each database was used as a variant database for analysis of variants on epitopes with VOE. The first database was the BCFtools variant database and the second was the SARS-CoV-2 Freebayes variant database.

VOE was coded with a Python script that locates variants on the epitope and calculates the sensitivity of these variants. The workflow for VOE is shown in Fig. 2. It uses BLAST (*Camacho et al., 2009*) tools that are publicly available in Bioconda on Unix/Linux platforms.

First, the nucleotide position of the epitope sequence from tBLASTn is generated using the default $E$-value ($E$-value < 10) (*Camacho et al., 2009*) (%identity = 100) and compared with the CDS in the SARS -CoV-2 database. The first hit from the tBLASTn result is selected. The reference database was downloaded from CDS nucleotide Wuhan-Hu-1 (NCBI Accession: NC_045512.2).

Second, all nucleotide positions of the epitope in the SARS-CoV-2 genome are calculated using the database according to the formula: nucleotide position in the gene from the tBLASTn result + nucleotide position in the genome from the database + 1.

Third, all variants on an epitope with HIGH or MODERATE effects (*Cingolani et al., 2012*) are included in the information of each variant from the variant database because both effects are assumed to change in amino acid sequence. All values in each column of the SRA accession number from the variant database are stored in a dictionary variable, such as {SRR: [0,0,0,1]} (0 means a variant item was not found in the SRA escrow number, 1 means a variant was found).

Last, using the dictionary variables, sensitivity is calculated based on the hypothesis that if a HIGH or MODERATE impact variant is found on the epitope, a false negative result (FN) follows, whereas if no variant or a synonymous variant, or both, are found on the epitope, a true positive result (TP) follows. If any of the values in the columns is 1, then

**Table 1  Modified variant database example.** POS_Genome is the position of the variant in the genome, CHROM is the name of the reference sequence, REF is the reference base, ALT means alternative alleles, INFO is generic information about this variant. FORMAT is an extensible list of fields for describing the samples. The rest columns are the number of the variant for each sample.

| POS | CHROM | REF | ALT | INFO | FORMAT | SRRX1 | SRRX2 | SRRX3 | SRRX4 | SRRX5 |
|-----|-------|-----|-----|------|--------|-------|-------|-------|-------|-------|
| 14 | NC_045512.2 | A | G | AC=2;AF=0.4;AN=5;NS=5…missense_variant\|MODERATE\|… | GT | 0 | 1 | 0 | 1 | 0 |
| 15 | NC_045512.2 | C | T | AC=1;AF=0.2;AN=5;NS=5…missense_variant\|MODERATE\|… | GT | 0 | 0 | 0 | 1 | 0 |
| 30 | NC_045512.2 | C | AAC | AC=3;AF=0.6;AN=5;NS=5…insertion\|HIGH\|… | GT | 0 | 0 | 1 | 1 | 1 |
| 20020 | NC_045512.2 | CCA | C | AC=2;AF=0.4;AN=5;NS=5…deletiong\|HIGH\| | GT | 1 | 0 | 1 | 0 | 0 |

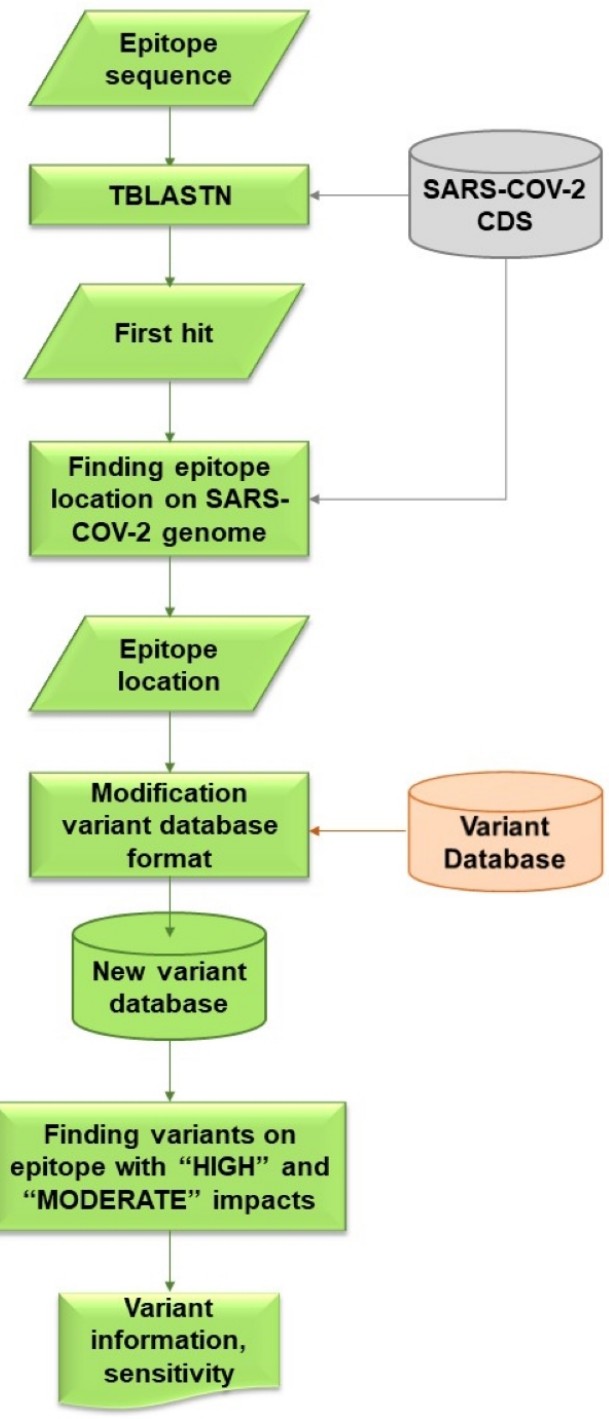

**Figure 2    Flowchart describing processes and steps of VOE.**

this SRA results in a FN. Conversely, if all of the values in the columns are 0, then this SRA results in a TP. The sensitivity of the epitope is calculated using the following formula (*Trevethan, 2017*).

$$Sensitivity = \frac{TP}{TP+FN}. \tag{1}$$

For example, suppose an epitope is located at nucleotide positions 10–30 and in this position interval, no variant is found for SRRX1, one variant is found for SRRX2, one variant is found for SRRX3, three variants are found for SRRX4, and one variant is found for SRRX5. Then, TP = 1 and FN = 4, and sensitivity is 1/(1+4) or 0.2 and %sensitivity is 20%.

### Peptide-protein docking validation

Peptides of epitopes were modeled with PEP-FOLD 3 Server (*Lamiable et al., 2016*). Peptide-protein docking of epitopes with lower 90% sensitivity and the most frequently occurring variant against specific proteins were analyzes using AutoDocktools version 1.5.7 (*Morris et al., 2009*). An increase in binding energy for mutant docking will make less compact binding of the epitope.

## RESULTS

### Identification of the SARS-CoV-2 lineage and trimming

The sequence reads of selected 1,791 SRA files were downloaded and assembled as 1,659 consensus files, which were identified with 1,023 Omicron accession numbers. The quality of reads of the accession numbers were assessed, giving 1,011 accepted accession numbers.

### VOE variant analysis and sensitivity calculation

Seven epitopes including KLNDLCFTNV, ITLCFTLKRK, GGDGKMKD, RVQPTES, APGQTGK, DSKVGGNYN, and LKPFERD were obtained to evaluate the performance of the proposed VOE analysis tool. The epitopes KLNDLCFTNV, ITLCFTLKRK, RVQPTES, APGQTGK, DSKVGGNYN, and LKPFERD are highly antigenic for the development of an epitope-based peptide vaccine and the epitope GGDGKMKD was identified as ideal for serodiagnostic testing.

The tBLASTn results of all epitopes from the BCFtools variant database appear in Table 2.

The first epitope KLNDLCFTNV is located on the S gene at nucleotide positions 1,156 to 1,185 (amino acid positions 386 to 395 on the S gene) with an $E$-value of $4.81 \times 10^{-4}$. The second epitope ITLCFTLKRK is located at nucleotide positions 328 to 357 on the ORF7a gene (amino acid positions 110 to 119) with an $E$-value of 0.005. The third epitope RVQPTES is located at nucleotide positions 955 to 975 on the S gene (amino acid positions 319 to 325) with an $E$-value of 0.025. The fourth epitope APGQTGK is located at nucleotide positions 1,231 to 1,251 on the S gene (amino acid positions 411 to 417) with an $E$-value of 0.028. The fifth epitope DSKVGGNYN is located at nucleotide positions 1,324 to 1,350 on the S gene (amino acid positions 442 to 450) with an $E$-value of 0.003. The sixth epitope LKPFERD is located at nucleotide positions 1,381 to 1,401 on the S gene (amino acid

**Table 2  tBlastn results of the BCFtools variant database.**

| Epitope | Variant found | E-value | %Identity | Nucleotide position on gene | Nucleotide position on the genome |
|---|---|---|---|---|---|
| KLNDLCFTNV (*Can et al., 2020*) | No | $4.81 \times 10^{-4}$ | 100% | 1,156 to 1,185 (S386-395) | 22,718 to 22,747 |
| ITLCFTLKRK (*Can et al., 2020*) | No | 0.005 | 100% | 328 to 357 (ORF7a110-119) | 27,722 to 27,750 |
| RVQPTES (*Ferreira et al., 2021*) | Yes | 0.025 | 100% | 955 to 975 (S319-325) | 22,517 to 22,537 |
| APGQTGK (*Ferreira et al., 2021*) | Yes | 0.028 | 100% | 1,231 to 1,251 (S411-417) | 22,793 to 22,813 |
| DSKVGGNYN (*Ferreira et al., 2021*) | Yes | 0.003 | 100% | 1,324 to 1,350 (S442-450) | 22,886 to 22,912 |
| LKPFERD (*Ferreira et al., 2021*) | No | 0.008 | 100% | 1,381 to 1,401 (S461-467) | 22,943 to 22,963 |
| GGDGKMKD (*Can et al., 2020*) | Yes | 0.12 | 100% | 286 to 309 (N96-103) | 28,559 to 28,582 |

positions 461 to 467) with an *E*-value of 0.008. The last epitope GGDGKMKD is located at nucleotide positions 286 to 309 on the N gene (amino acid positions 96 to 103) an *E*-value of 0.12. The % identity of all epitopes was 100%.

The variant information for all epitopes from the BCFtools variant database appear in Table 3. For the epitopes KLNDLCFTNV (S386-395) and LKPFERD (S461-467) on the S gene and the epitope ITLCFTLKRK (ORF7a110-119) on the ORF7a gene, no variants with two effects (HIGH or MODERATE) were found, resulting in a sensitivity of 100%. In contrast, for the epitopes RVQPTES (S319-325), APGQTGK (S411-417), DSKVGGNYN (S442-450) on the S gene, and GGDGKMKD (N96-103) on the N gene, variants were found with two effects, and one (p.Thr323Ile), three (p.Lys417Thr, p.Lys417Met, p.Lys417Asn), seven (p.Lys444Thr, p.Lys444Asn, p.Val445Leu, p.Val445Ala, p.Gly446Ser, p.Gly446Asp, p.Asn450Asp), and four (p.Gly96Cys, p.Gly96Val, p.Gly97Cys, p.Gly99Cys) missense variants, respectively, giving respective sensitivities of 99.9011%, 28.4866%, 87.7349%, and 99.6044%.

The tBLASTn results of all epitopes from the SARS-CoV-2-Freebayes variant database appear in Table 4. The first epitope KLNDLCFTNV is located at nucleotide positions 1,156 to 1,185 on the S gene (amino acid positions 386 to 395 on the S gene) with an *E*-value of $4.81 \times 10^{-4}$. The second epitope ITLCFTLKRK is located at nucleotide positions 328 to 357 on the ORF7a gene (amino acid positions 110 to 119) with an *E*-value of 0.005. The third epitope RVQPTES is located at nucleotide positions 955 to 975 on the S gene (amino acid positions 319 to 325) with an *E*-value of 0.025. The fourth epitope APGQTGK is located at nucleotide positions 1,231 to 1,251 on the S gene (amino acid positions 411 to 417) with an *E*-value of 0.028. The fifth epitope DSKVGGNYN is located at nucleotide positions 1,324 to 1,350 on the S gene (amino acid positions 442 to 450) with an *E*-value of 0.003. The sixth epitope LKPFERD is located at nucleotide positions 1,381 to 1,401 on the S gene (amino acid positions 461 to 467) with an *E*-value of 0.008. The last epitope
**Table 3  Variant information of the seven epitopes from the BCFtools variant database.** The POS_Genome column in Table 3 lists the nucleotide position of each variant in the genome, the ALT column shows the changed nucleotide sequence of each variant. AA_change is the changed amino acid sequence of each variant, AC is the allele count of samples, NS is the number of samples, AF is the allele frequency of each allele, and %Chance is the possibility of each identified variant (AC/NS*100).

| Epitope | POS _Genome | Type | ALT | AA _change | AC | NS | AF | Chance (%) | Sensitivity (%) |
|---|---|---|---|---|---|---|---|---|---|
| KLNDLCFTNV (S386-395) | | | The variant was not found | | | | | | |
| ITLCFTLKRK (ORF7a110-119) | | | | | | | | | 100 |
| RVQPTES (S319-325) | 22,530 | missense_variant | c.968C>T | p.Thr323Ile | 1 | 1,011 | 0.001 | 0.0989 | 99.9011 |
| APGQTGK (S411-417) | 22,812 | | c.1250A>C | p.Lys417Thr | 1 | 1,011 | 0.001 | 0.0989 | |
| | 22,812 | missense_variant | c.1250A>T | p.Lys417Met | 1 | 1,011 | 0.001 | 0.0989 | 28.4866 |
| | 22,813 | | c.1251G>T | p.Lys417Asn | 723 | 1,011 | 0.7151 | 71.5134 | |
| DSKVGGNYN (S442-450) | 22,893 | | c.1331A>C | p.Lys444Thr | 19 | 1,011 | 0.0188 | 1.8793 | |
| | 22,894 | missense_variant | c.1332G>T | p.Lys444Asn | 1 | 1,011 | 0.001 | 0.0989 | 87.7349 |
| | 22,895 | | c.1333G>C | p.Val445Leu | 1 | 1,011 | 0.001 | 0.0989 | |
| | 22,896 | | c.1334T>C | p.Val445Ala | 3 | 1,011 | 0.003 | 0.2967 | |
| | 22,898 | | c.1336G>A | p.Gly446Ser | 99 | 1,011 | 0.0979 | 9.7923 | |
| | 22,899 | | c.1337G>A | p.Gly446Asp | 1 | 1,011 | 0.001 | 0.0989 | |
| | 22,910 | | c.1348A>G | p.Asn450Asp | 2 | 1,011 | 0.002 | 0.1978 | |
| LKPFERD (S461-467) | | | Variant was not found | | | | | | 100 |
| GGDGKMKD (N96-103) | 28,559 | | c.286G>T | p.Gly96Cys | 1 | 1,011 | 0.001 | 0.0989 | |
| | 28,560 | missense_varian | c.287G>T | p.Gly96Val | 1 | 1,011 | 0.001 | 0.0989 | 99.6044 |
| | 28,562 | | c.289G>T | p.Gly97Cys | 1 | 1,011 | 0.001 | 0.0989 | |
| | 28,568 | | c.295G>T | p.Gly99Cys | 1 | 1,011 | 0.001 | 0.0989 | |

GGDGKMKD is located at nucleotide positions 286 to 309 on the N gene (amino acid positions 96 to 103) an *E*-value of 0.12. The % identity of all epitopes was 100%.

The variant information of all epitopes from the SARS-CoV-2-Freebayes variant database was presented in Table 5. For the epitope KLNDLCFTNV (S386-395), variants with two effects, three missense variants (p.Asn388Tyr, p.Asp389Val, and p.Leu390Pro), and one frameshift variant (p.Thr393fs) were found on the S gene. For the epitope ITLCFTLKRK (ORF7a110-119) on the ORF7a variant, five missense variants (p.Ile110Thr, p.Phe114Val, p.Thr115Ile, p.Leu116Ile, and p.Lys119Arg) were found. For the epitope RVQPTES (S319-325) on the S gene variant, one stop-gain (p.Arg319*), one frameshift (p.Pro322fs), and two missense variants (p.Val320Phe, and p.Thr323Ile) were found. For the epitopes APGQTGK (S411-417), DSKVGGNYN (S442:450), and LKPFERD (S461-467) on the S gene variant, seven (p.Gln414Lys, p.Lys417Asn, p.Lys417His, p.Lys417Asn, p.Lys417Thr, p.Lys417Ile, and p.Lys417Asn), 10 (p.Lys444Thr, p.Lys444Asn, p.ValGly445ProSer, p.Gly446Ser, p.Val445Ala, p.ValGly445AlaSer, p.Gly446Ser, p.Gly446Asp, p.Asn448Ile, and p.Asn450Asp) and two missense variants (p.Pro463Leu, and p.Asp467Val) were found, respectively. For the epitope GGDGKMKD (N96-103) on the N gene variant, four missense variants (p.Gly96Cys, p.Gly96Val, p.Gly97Cys, and p.Gly99Cys) were found.

**Table 4** tBlastn results of the Freebayes variant database.

| Epitope | Variant found | E-value | %Identity | Nucleotide position on gene | Nucleotide position on the genome |
|---|---|---|---|---|---|
| KLNDLCFTNV (*Can et al., 2020*) | Yes | $4.81 \times 10^{-4}$ | 100% | 1,156 to 1,185 (S386:395) | 22,718 to 22,747 |
| ITLCFTLKRK (*Can et al., 2020*) | Yes | 0.005 | 100% | 328 to 357 (ORF7a110:119) | 27,722 to 27,750 |
| RVQPTES (*Ferreira et al., 2021*) | Yes | 0.025 | 100% | 955 to 975 (S319:325) | 22,517 to 22,537 |
| APGQTGK (*Ferreira et al., 2021*) | Yes | 0.028 | 100% | 1,231 to 1,251 (S411:417) | 22,793 to 22,813 |
| DSKVGGNYN (*Ferreira et al., 2021*) | Yes | 0.003 | 100% | 1,324 to 1,350 (S442:450) | 22,886 to 22,912 |
| LKPFERD (*Ferreira et al., 2021*) | Yes | 0.008 | 100% | 1,381 to 1,401 (S461:467) | 22,943 to 22,963 |
| GGDGKMKD (*Can et al., 2020*) | Yes | 0.12 | 100% | 286 to 309 (N96:103) | 28,559 to 28,582 |

Lastly, the respective % sensitivities of all epitopes were 99.5054%, 99.5054%, 99.6044%, 6.8249%, 71.1177%, 99.8022%, and 99.6044%.

**Peptide-protein docking validation**

Epitopes with low sensitivity (<90%) could be considered moderate to low binders and were further investigated. The 3D structures of the B cell receptor (BCR) (ID: 5IFH) (*Rantam et al., 2021*) were obtained from the RCSB Protein Data Bank. The wild-type epitope (APGQTGK) and the mutant epitope (APGQTGN) were docked with the BCR shown in Figs. 3A and 3B, respectively. The estimated binding energy was −2.56 kcal/mol for the wild-type docking and −2.02 kcal/mol for the mutant docking. The wild-type epitope (DSKVGGNYN) and mutant epitope (DSKVSGNYN) were docked with the BCR shown in Figs. 4A and 4B, respectively. The estimated binding energy was −1.69 kcal/mol for wild-type docking and +3.58 kcal/mol for mutant docking.

## DISCUSSION

In previous predictions of human leukocyte antigen (HLA) class I and II epitopes, the binding of HLA class I and II molecules with pathogen peptides was an important trigger of T cell activity and other components of the adaptive immune response (*Can et al., 2020*). The epitope KLNDLCFTNV on the S gene was investigated in HLA class I epitope prediction. The result showed that, of 1,312 HLA class I and II epitopes, only 125 (9%) were modified in the Omicron variant. The KLNDLCFTNV epitope was not altered by the Omicron variant definition (*Centre of Disease Prevention E, 2021*; *Ukhsa, 2022*; *Chen et al., 2022*), and both cytotoxic and helper cellular immune protection elicited by currently licensed vaccines were not affected by the Omicron variant of SARS-CoV-2. Moreover, the epitopes KLNDLCFTNV and ITLCFTLKRK were recommended for the development of an epitope-based vaccine. In addition, the potential of KLNDLCFTNV to generate a

**Table 5 Variant information of seven epitopes from the SARS-CoV-2 Freebayes variant database.** The column headed POS_Genome lists the nucleotide position of each variant in the genome. The ALT column lists the changed nucleotide sequence of each variant. AA_change is the changed amino acid sequence of each variant, AC is the allele count of samples, NS is the number of samples, AF is the allele frequency of each allele, and %Chance is the possibility of each identified variant (AC/NS*100).

| Epitope | POS_Genome | Type | ALT | AA_change | AC | NS | AF | Chance (%) | Sensitivity (%) |
|---|---|---|---|---|---|---|---|---|---|
| KLNDLCFTNV (S386-395) | 22,724 | | c.1162A>T | p.Asn388Tyr | 1 | 1,011 | 0.001 | 0.0989 | |
| | 22,728 | missense_variant | c.1166A>T | p.Asp389Val | 2 | 1,011 | 0.002 | 0.0198 | 99.5054 |
| | 22,731 | | c.1169T>C | p.Leu390Pro | 1 | 1,011 | 0.001 | 0.0989 | |
| | 22,735 | frameshift_variant | c.1176dupT | p.Thr393fs | 1 | 1,011 | 0.001 | 0.0989 | |
| ITLCFTLKRK (ORF7a110-119) | 27,722 | | c.329T > C | p.Ile110Thr | 1 | 1,011 | 0.001 | 0.0989 | |
| | 27,733 | | c.340T>G | p.Phe114Val | 1 | 1,011 | 0.001 | 0.0989 | |
| | 27,737 | missense_variant | c.344C>T | p.Thr115Ile | 1 | 1,011 | 0.001 | 0.0989 | 99.5054 |
| | 27,739 | | c.346C>A | p.Leu116Ile | 1 | 1,011 | 0.001 | 0.0989 | |
| | 27,749 | | c.356A >G | p.Lys119Arg | 1 | 1,011 | 0.001 | 0.0989 | |
| RVQPTES (S319-325) | 22,517 | stop_gained | c.955A>T | p.Arg319* | 1 | 1,011 | 0.001 | 0.0989 | |
| | 22,520 | missense_variant | c.958G>T | p.Val320Phe | 1 | 1,011 | 0.001 | 0.0989 | 99.6044 |
| | 22,525 | frameshift_variant | c.965delC | p.Pro322fs | 1 | 1,011 | 0.001 | 0.0989 | |
| | 22,530 | missense_variant | c.968C>T | p.Thr323Ile | 1 | 1,011 | 0.001 | 0.0989 | |
| APGQTGK (S411-417) | 22,802 | | c.1240C>A | p.Gln414Lys | 1 | 1,011 | 0.001 | 0.0989 | |
| | | | c.1251G>T | p.Lys417Asn | 1 | 1,011 | 0.001 | 0.0989 | |
| | 22,811 | | c.1249_1251delAAGinsCAT | p.Lys417His | 1 | 1,011 | 0.001 | 0.0989 | |
| | | missense_variant | c.1251G>T | p.Lys417Asn | 1 | 1,011 | 0.001 | 0.0989 | 6.8249 |
| | 22,812 | | c.1250_1251delAGinsCT | p.Lys417Thr | 1 | 1,011 | 0.001 | 0.0989 | |
| | 22,812 | | c.1250_1251delAGinsTT | p.Lys417Ile | 1 | 1,011 | 0.001 | 0.0989 | |
| | 22,813 | | c.1251G>T | p.Lys417Asn | 938 | 1,011 | 0.9278 | 92.7794 | |
| DSKVGGNYN (S442-450) | 22,893 | | c.1331A>C | p.Lys444Thr | 42 | 1,011 | 0.0415 | 4.1543 | |
| | 22,894 | | c.1332G>T | p.Lys444Asn | 1 | 1011 | 0.001 | 0.0989 | |
| | 22,895 | | c.1333_1336delGTTGinsCCTA | p.ValGly445ProSer | 19 | 1,011 | 0.0188 | 1.8793 | |
| | | missense_varian | c.1336G>A | p.Gly446Ser | 3 | 1,011 | 0.003 | 0.2967 | 71.1177 |
| | 22,896 | | c.1334T>C | p.Val445Ala | 4 | 1,011 | 0.004 | 0.3956 | |
| | 22,896 | | c.1334_1336delTTGinsCTA | p.ValGly445AlaSer | 1 | 1,011 | 0.001 | 0.0989 | |
| | 22,898 | | c.1336G>A | p.Gly446Ser | 222 | 1,011 | 0.2196 | 21.9585 | |
| | 22,899 | | c.1337G>A | p.Gly446Asp | 1 | 1,011 | 0.001 | 0.0989 | |
| | 22,905 | | c.1343A>T | p.Asn448Ile | 1 | 1,011 | 0.001 | 0.0989 | |
| | 22,910 | | c.1348A>G | p.Asn450Asp | 3 | 1,011 | 0.003 | 0.2967 | |
| LKPFERD (S461-467) | 22,950 | missense_variant | c.1388C>T | p.Pro463Leu | 1 | 1,011 | 0.001 | 0.0989 | 99.8022 |
| | 22,962 | | c.1400A>T | p.Asp467Val | 1 | 1,011 | 0.001 | 0.0989 | |
| GGDGKMKD (N96-103) | 28,559 | | c.286G>T | p.Gly96Cys | 1 | 1,011 | 0.001 | 0.0989 | |
| | 28,560 | | c.287G>T | p.Gly96Val | 1 | 1,011 | 0.001 | 0.0989 | |
| | 28,562 | missense_variant | c.289G>T | p.Gly97Cys | 1 | 1,011 | 0.001 | 0.0989 | 99.6044 |
| | 28,568 | | c.295G>T | p.Gly99Cys | 1 | 1,011 | 0.001 | 0.0989 | |

**Notes.**

An asterisk (*) indicates stop codon.

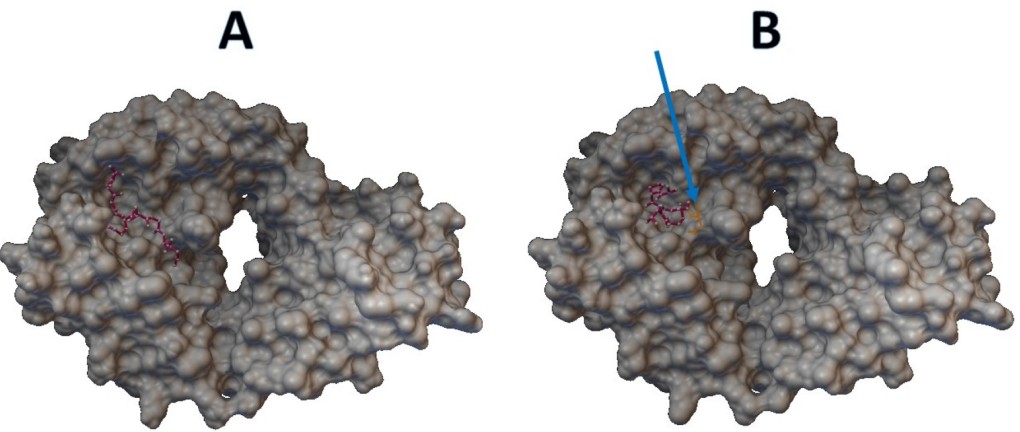

**Figure 3** **Molecular docking between epitope APGQTGK and B cell receptor.** (A) Wild-type epitope. (B) Mutant epitope.

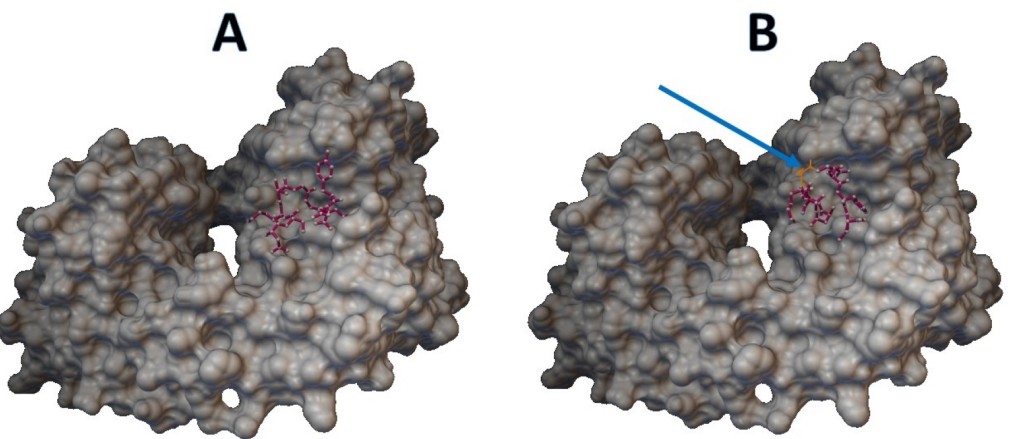

**Figure 4** **Molecular docking between epitope DSKVGGNYN and B cell receptor.** (A) Wild-type epitope. (B) Mutant epitope.

vaccine was demonstrated and confirmed *in vivo* (*Shen et al., 2022*). Our results showed that the sensitivity of both epitopes was 100% when calculated from the BCFtools variant database and 99.5054% when calculated from the SARS-CoV-2- Freebayes variant database (Tables 3 and 5). Therefore, both epitopes could be strongly recommended for use in the development of an epitope-based vaccine.

In a previous study on single mutations affecting viral escape antibodies (*Doud, Lee & Bloom, 2018*), it was found that some point mutations can affect antibody binding. In addition, for peptide-based vaccines, some epitope prediction, such as the B-cell T-cell response, is immunoinformatically required (*Oli et al., 2020*; *Ramana & Mehla, 2020*; *Lu et al., 2021a*). The epitopes RVQPTES, APGQTGK, DSKVGGNYN, and LKPFERD on the S gene were B-cell epitopes for which nonsynonymous variants were found in both variant

databases, except for LKPFERD, for which nonsynonymous variants were found only in the BCFtools variant database. Also, epitopes APGQTGK, DSKVGGNYN and LKPFERD have been proposed for use in in the wet lab for vaccine design (*Lu et al., 2021b*). Specifically, the epitope APGQTGK on the S gene was identified by the Omicron variant definition (*Centre of Disease Prevention E, 2021*) at K417N or p.Lys417Asn with a sensitivity of 28.4866% in the BCFtools database and 6.8249% in the SARS-CoV-2 Freebayes database. In both variant databases, the epitope DSKVGGNYN on the S gene was modified by the Omicron variant definition (*Centre of Disease Prevention E, 2021*) at G446S with respective sensitivities of 87.7349% and 71.1177%. The peptide-protein docking of the mutant epitopes APGQTGK and DSKVGGNYN also leads to an increase in binding energy, resulting in a less compact binding of the epitopes. Therefore, the epitopes APGQTGK and DSKVGGNYN could be further considered in terms of variant impact on the epitope-based vaccine development. In contrast, the RVQPTES and LKPFERD epitopes on the S gene were not modified by the Omicron variant definition (*Centre of Disease Prevention E, 2021*). In addition, the sensitivity of the epitopes RVQPTES and LKPFERD was almost 100% in both variant databases.

An earlier report of a rapid diagnostic test for SARS-CoV-2 indicated that novel viral mutations can directly alter the genomic sequence detected by molecular RDTs. Ag-RDTs recognize epitopes on surface proteins (mostly the nucleocapsid), and their performance depends more on protein structure and confirmation than on individual genomic mutations (*Drain, 2022*). GGDGKMKD is a sub-sequence of the nucleocapsid protein recommended as ideal for serodiagnostic test development. Our results (Tables 3 and 5) showed four nonsynonymous variants of the epitope GGDGKMKD on the N gene from both variant databases, and sensitivity is 99.6044%. Therefore, its sensitivity can be assessed by comparison with an NAAT such as rRT-PCR. Moreover, the data in Tables 3 and 5 can be used in protein modeling for analysis of the impact on each variant.

Since VOE was developed using the Python programming language, the program can run on either Windows or Linux platforms. Moreover, VOE is easy to use, and processing is completed within one minute. However, the limitation of VOE is that the query sequence must contain more than five amino acids due to a tBLASTn condition.

## CONCLUSIONS

In the development of epitope-based vaccines and Ag-RDTs, mutation is one of the most important factors since mutations can reduce the binding affinity between the epitope and protein, resulting in less sensitive Ag-RDTs and less effective vaccines. VOE is a Python script that allows users to quickly identify variants and calculate the sensitivity of epitopes. The results from VOE can be used as data for identifying suitable epitopes. In the future, results from VOE should be studied in terms of protein structure and conformation. In addition, all epitopes should be docked with HLAs and/or TLRs proteins to ensure that the epitopes actually elicit an immune response under the simulated biochemical reaction conditions. The BCFtools pipeline and the SARS-CoV-2 Freebayes pipeline can be used to create new variant databases from new variants of SARS-CoV-2 or new SRA data, and VOE can be used to analyze other epitopes.

## ACKNOWLEDGEMENTS

We thank Mr. Thomas Coyne for language proofreading.

### Funding

This research was supported by the National Science, Research and Innovation Fund (NSRF) and Prince of Songkla University (Grant No. SCI6701310S). The funders had no role in study design, data collection and analysis, decision to publish, or preparation of the manuscript.

### Grant Disclosures

The following grant information was disclosed by the authors:
National Science, Research and Innovation Fund (NSRF).
Prince of Songkla University: SCI6701310S.

### Competing Interests

The authors declare there are no competing interests.

### Author Contributions

- Danusorn Lee conceived and designed the experiments, performed the experiments, analyzed the data, prepared figures and/or tables, authored or reviewed drafts of the article, and approved the final draft.
- Unitsa Sangket conceived and designed the experiments, performed the experiments, analyzed the data, prepared figures and/or tables, authored or reviewed drafts of the article, and approved the final draft.

### Data Availability

The VOE source code, BCFtools_variant database, Freebayes_variant database and its manual are available at GitHub and Zenodo:

- https://github.com/lee99dn/SARS-CoV-2-VOE.
- lee99dn, & unitsa-sangket. (2024). lee99dn/SARS-CoV-2-VOE: first release version of VOE (V.1.0.0). Zenodo. https://doi.org/10.5281/zenodo.11197886.

The 1,659 SRA accession numbers are available in the Supplementary File.

### Supplemental Information

Supplemental information for this article can be found online at http://dx.doi.org/10.7717/peerj.17504#supplemental-information.

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
