# Peer review of "VOE: automated analysis of variant epitopes of SARS-CoV-2 for the development of diagnostic tests or vaccines for COVID-19"

_PeerJ, doi:10.7717/peerj.17504_

## Round 0.1 · original submission · Major Revisions

First, sorry for the delay in my decision; the third reviewer has requested to extend the review deadline but she/he has not submitted the comments; maybe we will send you additional comments when available.

Anyway, I made the decision based on the comments of the other two reviewers. Please read their comments carefully and revise the manuscript accordingly.

Reviewer 1 ·

Basic reporting

The flow of English languange is no problem. References, and context are sufficiently explained. However, almost all figures are provided in low resolution. The quality of the figures should be improved accordingly. This research also hypothezied that if a missense variant is
found on an epitope, epitope binding will be different. The result and discussion section could provide clear explanation on the hypothesis. More comments, please kindly check the annotated manuscript.

Experimental design

The research is totally aligned with the aims and scope of the journal. This research could fills and identified knowledge gap in terms of genomic surveillance of SARS-CoV-2 in the southern Thailand. Methods are sufficiently detailed. However, some aspects of the methodology will need further clarification in order to validate the findings. More comments, please kindly check the annotated manuscript.

Validity of the findings

In general, the findings could be considered valid. However, there are specific indicators and parameters that should be clarified in order to uphold the scientific rigor. More comments, please kindly check the annotated manuscript.

• BLAST e value cut off: The e value cutoff of BLAST above 10e-3 will introduce false positives
• Molecular docking validation: The molecular docking validation should be conducted with spike protein of Wuhan variant.
• Epitopes sensitivity: The sensitivity of below 90% could be considered moderate to low binders.
• Python script: The script could not be found in the supplementary material. Please kindly upload the script in the supplement. In the very least, kindly provide the pseudocodes of the program, so the algorithm could be replicated accordingly.

Additional comments

Discussion section would need to point out the need to do molecular simulation research for the future venue of this project. In the future, all the epitopes should be docked with HLAs and/or the TLRs protein, in order to ensure that those epitopes really elicit immune response in the simulated biochemical reaction settings. Many research already conducted for this regard, and could be checked in this link:
https://pubmed.ncbi.nlm.nih.gov/?term=molecular+docking+epitopes+sars-cov-2
More comments, please kindly check the annotated manuscript.

Annotated reviews are not available for download in order to protect the identity of reviewers who chose to remain anonymous.

·

Basic reporting

-

Experimental design

-

Validity of the findings

-

Additional comments

Firstly of all, this paper needs English editing.
Title: Okay.
Abstract: This section was well-written and easy to understand. In addition, the study's objective and the state of the art of the study is clear. Furthermore, the keywords should represent the study.
Introduction: Based on my review, I recommend that the authors should add various improvements, such as:
- Hypothesis and objectives.
- Add a statement or sentence about whether there are any similar studies that have been done before.
- Add the important issues.
Material and Methods: This section should be a more detailed protocol or cite the reference.
Results: Clear.
Discussion: Comprehensive.
Conclusion: Clear. However, it might be helpful if the authors mention the suggestion for forthcoming research.
References: The core references are acceptable, but I think need some improvements, such as:
> https://link.springer.com/chapter/10.1007/978-3-030-63761-3_47

---

## Round 0.2 · accepted · Accept

Since both reviewers recommend the acceptance of your revised manuscript, I am happy to recommend its acceptance to the section editor. Congratulations!

Reviewer 1 ·

Basic reporting

The authors have follow up my comments accordingly

Experimental design

The authors have follow up my comments accordingly

Validity of the findings

The authors have follow up my comments accordingly

Additional comments

The authors have follow up my comments accordingly

·

Basic reporting

Clear

Experimental design

Clear

Validity of the findings

Clear